# Edge Magnetism in MoS_2_ Nanoribbons: Insights from a Simple One-Dimensional Model

**DOI:** 10.3390/nano13243086

**Published:** 2023-12-05

**Authors:** Pauline Castenetto, Philippe Lambin, Péter Vancsó

**Affiliations:** 1Department of Physics, University of Namur, 5000 Namur, Belgium; pauline.castenetto@unamur.be; 2Institut Supérieur Pédagogique, Bukavu P.O. Box 854, Democratic Republic of the Congo; 3Institute of Technical Physics and Materials Science, Center for Energy Research, 1121 Budapest, Hungary; vancso.peter@ek.hun-ren.hu

**Keywords:** DFT band structure, tight-binding calculations, Hubbard Hamiltonian, 2D materials

## Abstract

Edge magnetism in zigzag nanoribbons of monolayer MoS${}_{2}$ has been investigated with both density functional theory and a tight-binding plus Hubbard (TB+*U*) Hamiltonian. Both methods revealed that one band crossing the Fermi level is more strongly influenced by spin polarization than any other bands. This band originates from states localized on the sulfur edge of the nanoribbon. Its dispersion closely resembles that of the energy branch obtained in a linear chain of atoms with first-neighbor interaction. By exploiting this resemblance, a toy model has been designed to study the energetics of different spin configurations of the nanoribbon edge.

## 1. Introduction

Transition metal dichalcogenide (TMDC) monolayers have many properties [1,2] that make them suitable for applications in nanoelectronics, photonics, spintronics, and other fields [3,4]. Nanoribbons of these monolayers offer an even larger spectrum of properties due to the presence of edges [5].

As first predicted in the case of graphene [6], electronic states can be strongly localized on zigzag edges of 2D materials having a honeycomb lattice. In the case of graphene zigzag nanoribbons, the edge states correspond to flat portions of two bands approaching asymptotically the charge neutrality level from above and below at the X point of the Brillouin zone. The extension of the flat part increases with the width of the nanoribbon. This peculiar band structure is unstable with respect to spin polarization [7]: the two flat band portions are shifted above and below the charge neutrality level by ferromagnetic spin ordering of the electrons that occupy the edge states. According to the second Lieb theorem [8], the spins on both edges have opposite orientations.

Zigzag MoS${}_{2}$ nanoribbons also have metallic states localized either at their Mo edge or their S edge [9,10]. Whereas the interior of the nanoribbon is a semiconductor, the edge states give rise to two to four dispersive bands crossing the Fermi level. The shape and number of these metallic bands depend on the Hamiltonian used and the edge geometry [11]. In the simplest case, one can preserve the primitive lattice constant of the nanoribbon by saturating every Mo atom at the Mo edge with a sulfur dimer [10,12]. The atomic structure just discussed is depicted in Figure 1. This experimentally observed geometry has been adopted in the present work. For that geometry, spin polarization takes place at the S edge and no magnetism is present at the Mo edge [12,13] (see Figure 1), which is also the case for a (2×1) reconstruction of both edges [14]. Spin-polarized band structure calculations of zigzag MoS${}_{2}$ nanoribbons, whatever the particular edge structure used, confirm that the magnetism first observed experimentally in edge-oriented nanosheets [15] is due to edge states. Since then, more and more experimental evidence of magnetism produced by electrons localized at zigzag edges of TMDC nanosheets has appeared in the literature [16,17,18,19].

An interesting observation from DFT band-structure calculations is the fact that a branch with strong sulfur-edge character that crosses the Fermi level is strongly responsive to spin polarization, whereas the other branches are virtually independent of the spin orientation [13]. This responsive branch is called the “magnetic band” here. Its component with spin down is fully located below the Fermi level, while the spin-up component is filled up at 30%. This band is responsible for a magnetic moment of 0.7 $\mu_{B}$ per one-dimensional unit cell of the nanoribbon when one Bohr magneton $\mu_{B}$ is attributed to each electron. When spin polarization is switched off, the magnetic band is weakly dispersive, as shown in Section 2. Anticipating the results of Section 2.2, the dispersion of the very same band described with a tight-binding Hamiltonian is a cosine-like function similar to what is obtained in calculating the band structure of a linear chain of atoms. By adding a Hubbard term to the on-site energies of the tight-binding Hamiltonian (TB+*U* Hamiltonian), the spin-degeneracy of the magnetic band is lifted, its two components being simply shifted from each other in energy. Like in DFT calculations, the other bands remain mainly degenerate in spin.

The observation just described led us to pay specific attention to this magnetic band, while forgetting all the other branches of the nanoribbon. The problem is thereby reduced to that of a linear chain of atoms described by a Hubbard model. The full TB+*U* calculation having been performed in the mean-field approximation, by neglecting all correlation effects, the linear chain will be treated at the same level of approximation. This toy model allows us to understand better the results of the full calculations, more specifically, the energetics of the magnetic Bloch domains. In addition to making the understanding easier, the linear-chain model does explain the magnetic properties of the S edge of zigzag TMDC nanoribbons. As demonstrated below, the ground state of the one-dimensional model is ferromagnetic, as predicted by full calculations for the nanoribbon. Based on this, the central motivation of the work was to estimate the excitation energies of different arrangements of the magnetic moments, such as antiferromagnetic order and reversed-spin domains of various sizes that are not accessible to full calculations. The main goal was to analyze the robustness of the ferromagnetic order. In summary, the results described in this paper usefully complete the picture we have been able to sketch in Ref. [13] and open a new pathway for the investigation of 1D edge magnetism in TMDC nanoribbons.

## 2. Methods

### 2.1. DFT Calculations

Here we recall the main steps of the DFT and TB band structure calculations [13] that are necessary to understand the one-dimensional (atomic chain) model presented in this paper. As the goal was to correctly set our tight-binding parameters and crystal structures, as well as ensure correct orbital dependency, we have compared the band structures obtained by tight-binding with ones obtained by density functional theory (DFT). The latter were computed with the Vienna *ab initio* simulation package (VASP) [20] using the projector augmented wave (PAW) method [21]. The exchange-correlation functional chosen is the Perdew–Burke–Ernzerhof generalized gradient approximation (GGA-PBE) [22]. The band structure was calculated with a plane-wave cutoff set at 500 eV and the Brillouin zone was sampled on a (12 × 2 × 1) Monkhorst–Pack mesh of *k* points [23]. The convergence criterion for forces was set to 0.01 eV/Åduring geometry optimization.

After the relaxation of the geometry, the S-S bond length of the dimer localized on the Mo-terminated edge of the zigzag nanoribbon was found to be 1.99 Å, which differs notably from both the in-plane (3.18 Å) and out-of-plane (3.13 Å) S-S bond lengths.

Our DFT calculations were compared with previous results, even on different edge geometries of MoS${}_{2}$. A perfect agreement was found, e.g., with Ref. [12]. As a general remark, all the calculations presented in this paper were performed at 0 K.

### 2.2. TB Formalism

We used a tight-binding (TB) model to calculate the band structures of MoS${}_{2}$. This method is similar to the LCAOs (linear combination of atomic orbitals) frequently used in quantum chemistry. The main assumption of this model is that the orbitals are localized and orthogonal, like Wannier functions of the assumed crystalline material. In this formalism, the allowed energies of the Schrödinger equation are the eigenvalues of the matrix whose elements are
(1)$$H_{i\lambda ,j\mu }=\left(\epsilon_{i\lambda }\delta_{\lambda \mu }\right)\delta_{ij}+\beta_{ij}^{\lambda \mu }$$
where the indices i,j refer to atomic sites, the Greek letters λ,μ designate *s*, *p*, or *d* orbitals, $\epsilon_{i\lambda }$ is the on-site energy, $\beta_{ij}^{\lambda \mu }$ the hopping integral, and *H* is a Hermitian matrix. The approach used in this paper is the Slater–Koster one, which consists in considering the different parameters of the tight-binding model as free parameters that can be adjusted to obtain the best possible band structure. This has the advantage of being fast to calculate numerically and of being applicable in situations for which Bloch’s theorem is not valid.

For MoS${}_{2}$, our model considers an orthogonal basis made of five orbitals ($4d_{xy}$, $4d_{yz}$, $4d_{xz}$, $4d_{x^{2}-y^{2}}$, and $4d_{3z^{2}-r^{2}}$) for each molybdenum atom and three orbitals for each sulfur atom ($3p_{x},3p_{y},3p_{z}$). We first used the TB parameters (hopping terms and on-site energies) found in Ref. [24] and Ref. [25]. Since these parameters were fitted to infinite sheets of MoS${}_{2}$, they fail to describe accurately the band structure of a nanoribbon, as the edges are not taken into account. Thus, the first step was to modify the on-site energy parameters of the edge atoms in order to quantitatively reproduce the DFT band structure calculations.

As the goal was to be as close as possible to what could be found experimentally, we focused on MoS${}_{2}$ nanoribbons with zigzag edges. One edge is S-terminated and the other one is Mo-terminated and passivated with S dimers, for this is the most stable configuration according to both theoretical predictions and experimental observations [26]. The crystal structure of the nanoribbon is shown in Figure 1.

To reproduce the shape of the band structure and mid-gap states with strong edge character in the DFT calculations, it was necessary to describe individually the atoms at the edges as well as the S dimer with the fine-tuned parameters shown in Table 1. The orbital dependency of each mid-gap state in our parameterized TB model is correctly reproduced compared to the DFT results. The width of the nanoribbon (the number of zigzag lines across the nanoribbon) was fixed at six for our parameterization. This turns out to be sufficient to avoid electronic interaction between the two edges [13]. Therefore, our parameterization is valid to describe wider nanoribbons.

Four mid-gap states can be seen in Figure 2: three of them cross the Fermi level, resulting in the existence of metallic states. The strong localization of these metallic states at the edges of the nanoribbon is clearly revealed by charge density plots around the Fermi level displayed in fig. 2(d) of Ref. [13]. The blue and green branches are states localized on the S and Mo atoms at the S edge, whereas the red and yellow states are coming from the S dimers and Mo atoms at the Mo edge.

After this first step, we had to take into account the magnetic properties of the zigzag nanoribbon. For this, we switched to a Hubbard-TB approach in the grand-canonical ensemble and interaction terms $U_{Mo}$ and $U_{S}$ corresponding to the five Mo and three S orbitals:(2)$$H=\sum_{\langle ij\rangle s}t_{ij}{\hat{c}}_{i,s}^{†}{\hat{c}}_{j,s}+U_{Mo}\sum_{i\in Mo}{\hat{n}}_{i↑}{\hat{n}}_{i↓}+U_{S}\sum_{i\in S}{\hat{n}}_{i↑}{\hat{n}}_{i↓}+\sum_{i,s}\left(\epsilon_{i}-\mu \right){\hat{n}}_{is}$$
where $t_{ij}$ are the hopping parameters, ${\hat{c}}_{i,s}$ annihilates a fermion at site *i* with spin *s* (and ${\hat{c}}_{i,s}^{†}$ creates one), $\epsilon_{i}$ are the on-site energy parameters, ${\hat{n}}_{i,s}$ the particle-number operator, and μ the chemical potential. For simplicity, the orbital index is ignored here, but the calculations were indeed performed with five and three orbitals per Mo and S atoms, respectively. Since we are studying large systems in this paper, we apply the standard mean-field decoupling of the Hubbard terms: ${\hat{n}}_{i↑}{\hat{n}}_{i↓}\approx n_{i↑}{\hat{n}}_{i↓}+n_{i↓}{\hat{n}}_{i↑}-n_{i↑}n_{i↓}$, which yields an effective single-particle Hamiltonian that can be diagonalized either in *k*- or real-space. The average occupation numbers $n_{i,s}=\langle {\hat{n}}_{i,s}\rangle$ have to be adjusted self-consistently. Briefly stated, the algorithm is the following. A starting value is given to the average occupation numbers. The mean-field version of the Hamiltonian (Equation 2) can then be diagonalized and the local densities of states $\rho_{i,s}\left(E\right)$ are calculated for both spins on all the atoms and for all the orbitals. The total density of states follows from $\rho \left(E\right)=\sum_{i,s}\rho_{i,s}\left(E\right)$, where the sum over *i* represents the sum on all the orbitals of all the atoms. The Fermi energy $\epsilon_{F}$ and the occupation numbers are then calculated by the equations
(3a)$$\begin{array}{ccc} N_{e} & = & \int_{-\infty }^{\epsilon_{F}}\rho \left(E\right)dE \end{array}$$
(3b)$$\begin{array}{ccc} n_{i,s} & = & \int_{-\infty }^{\epsilon_{F}}\rho_{i,s}\left(E\right)dE \end{array}$$
where $N_{e}$ is the total number of electrons in the system. Once $\epsilon_{F}$ has been set through Equation ([Disp-formula FD3a-nanomaterials-13-03086]), new values are obtained for the $n_{i,s}$ set from Equation ([Disp-formula FD3b-nanomaterials-13-03086]). The initial values are corrected and the process is repeated until the $n_{i,s}$ values differ from the ones of the previous iteration by less than a given tolerance.

The Coulomb interactions related to the S and Mo atoms in Equation (Equation 2) were adjusted so as to best fit the spin-polarized DFT band structure of the nanoribbon with six unit cells across the thickness. The obtained values of the Hubbard parameters were $U_{S}$ = 1.7 eV and $U_{Mo}$ = 0.6 eV.

As visible in Figure 3, the effect of spin polarization is mainly noticeable on what we have called the “magnetic band”, coming from the S atoms at the S edge. The partially filled spin-up band is slightly shifted at higher energy, becoming less occupied, whereas the spin-down band is far down the Fermi level, leaving this band totally filled. There is virtually no magnetic moment on the Mo edge due to the S${}_{2}$ passivation of the Mo atoms. In the notation of Ref. [27], the magnetic ground state is the $F_{X}$ phase, meaning ferromagnetic ordering of the spins on the S edge of the nanoribbon. The magnetic structure of the nanoribbon is illustrated by spin-polarization plots presented in Figure 1.

At this stage of the discussion, it is important to point out that spin–orbit coupling (SOC) was not taken into account in the present work. Due to the absence of inversion symmetry in a monolayer TMDC, SOC lifts the spin degeneracy of the electronic bands and induces effects that can be felt in some physical phenomena [28]. The spin splitting depends on the character of the bands. It is important where the $d_{xy}$ and $d_{x^{2}-y^{2}}$ orbitals of the transition metal dominate the character of a band. By contrast, the splitting is very small for those bands that involve mainly the $d_{3z^{2}-r^{2}}$ orbital of the transition metal and/or the $p_{z}$ orbital of the chalgonenides [29]. We have performed spin-polarized DFT calculations for a MoS${}_{2}$ nanoribbon including SOC and we may confirm that the SOC has a very small effect on the magnetic band defined here above, because it has a dominant $3p_{z}$ character (see below). In the ribbon band structure, the splitting of this band is the same, with or without SOC, all along the ΓX line within 5 meV variations. This finding demonstrates that the splitting of the magnetic band is not due to spin–orbit coupling and comes, therefore, from electron–electron interactions. Briefly stated, the SOC does not modify the spin-polarization plot of Figure 1. This is fortunate, because including spin–orbit coupling in a TB+*U* Hamiltonian introduces new empirical parameters in the form of SOC constants [30], which would require validation for the nanoribbon geometry.

The advantage of the TB+*U* approach over the DFT method is in its much lower computing load. The Hamiltonian is a matrix whose size is 11 times the number of MoS${}_{2}$ units. In contrast to DFT, the diagonalization of the Hamiltonian for nanoribbons containing several hundreds of MoS${}_{2}$ units can be achieved in a reasonable computing time, including the self-consistent loop over the occupation numbers on the S and Mo orbitals.

In a previous paper [13], high-performance computing facilities allowed us to treat nanoribbons of 6×40 (width times length) units, thanks to the relative simplicity of the TB+*U* model. We have been able to show that, by applying randomly distributed Gaussian potentials, it was possible to play with the spin-texture of a MoS${}_{2}$ nanoribbon. Unlike zigzag graphene nanoribbons, MoS${}_{2}$ nanoribbons have weak magnetic coupling between the S atoms at their S edge. Therefore, using a positive potential stretching out on just a few atomic distances, it is possible to change the orientation of the edge spins, breaking locally the ferromagnetic ground state and creating domain walls. We demonstrated with a simple Gaussian potential that the width and the profile of the potential play a crucial role in the magnetic ground state of the system. In the case of overlapping regions where a potential is applied, the ferromagnetic order is preserved. However, a more systematic study of the effect of the potential on the magnetic properties requires simplified models that are less time consuming than the solution of the full TB+*U* Hamiltonian.

The shape of the magnetic band seems to be suitable for a one-dimensional chain model, which will allow us to have an interesting toy model to study, in a more systematic way, the effect of spin reversal on various scales and sites. In this way, the problem is oversimplified but we believe it preserves an essential part of the physics. The reason is that the magnetic band (blue curve in Figure 2) we focus all our attention on is weakly interacting with the other bands located nearby. Indeed, the black curves are bulk states having little weight at the edge atoms. The green branch comes from the same S edge as the blue one, but it is unoccupied and should not affect the properties of the system at 0 K. The red and yellow branches come from electronic states localized on the opposite edges of the nanoribbon. When the width of the nanoribbon is doubled, nothing changes in the relative position and shape of the mid-gap bands [13]. This means that, already with six units cells across the thickness, the edges of a zigzag nanoribbon are independent. It is not a strong approximation, therefore, to consider the magnetic band as being decoupled from the other bands.

## 3. Atomic Chain Model

### 3.1. Motivation

It was mentioned in the previous section that, compared to unpolarized calculations, spin polarization chiefly affects a single electronic band of the nanoribbon. This magnetic band is the one attributed to states localized on the S edge that crosses the Fermi level with positive slope. Its dispersion can merely be represented by the expression ϵ(k)=ϵ+βcos(ka), with *a* the one-dimensional period. This dispersion law is what is obtained in a one-dimensional atomic chain containing a single orbital on every site. Careful analysis of the DFT nanoribbon wavefunctions indicate that this orbital mixes the $3p_{y}$ and $3p_{z}$ orbitals of the two symmetric sulfur atoms of the S edge. The contribution of other orbitals, including those from Mo atoms, is very small. The mixed $3p_{y}$–$3p_{z}$ orbital, ∼40%–∼60% in proportion, is perpendicular to the edge direction, taken as *x* (see Figure 1). Accordingly, the small dispersion of the magnetic band observed in Figure 2 can be attributed to a weak π interaction between the mixed *p* orbitals on successive sulfur pairs. This π interaction is the hopping parameter β indicated in Figure 4.

The density of states $\rho (\epsilon_{F})$ at the Fermi energy of the nanoribbon is big due to the small dispersion of the magnetic band. Taking into account the Hubbard interaction $U_{S}$, one may anticipate that the Stoner criterion [31]$U_{S}\rho \left(\epsilon_{F}\right)>1$ is met; see below. As a consequence, the magnetic band is unstable with respect to spin splitting. In the spin-polarized band structure of Figure 3, indeed, the spin degeneracy is lifted, the band with spin ↓ shifts down, the one with spin ↑ still crosses the Fermi level, hence the ferromagnetic character of the S edge of the MoS${}_{2}$ nanoribbon.

In this section, we focus on this particular band and analyze its behavior in the context of the Hubbard model. The parameters of our toy model are ϵ and β for the dispersion law, and the number of electrons per spin $n_{0}$ contained in the band. The parameters fitted to the ab initio and full TB+*U* calculations are listed in Table 2. β is the quarter of the band dispersion between the Γ and the *X* points of the first Brillouin zone and ϵ is the average energy of the band. The effective *U* parameter was obtained by dividing by $2(1-n_{0})$ = 0.7 the splitting of 0.52 eV between the spin ↑ and spin↓ bands in the full TB+*U* calculations (Figure 3).

A great advantage of the one-dimensional model is its simplicity. Unlike the full TB+*U* calculations, there is no need for a supercell approach with periodic boundary conditions. Local densities of states in a chain containing several thousands sites can be calculated in a very efficient way, making it possible to study different kinds of disorder of the magnetic structure.

We have used the mean-field approximation, although the one-dimensional Hubbard model can be solved analytically for a periodic chain [32,33,34]. There are several reasons for keeping this approximation: (1) we want to remain as close as possible to the full nanoribbon TB+*U* calculations which were performed at the Hartree–Fock level; (2) the prediction of the properties of the one-dimensional periodic Hubbard model based on its analytical solution is complicated; (3) our main interest lies in non-periodic magnetic configurations involving a large number of sites, which makes the exact diagonalization of the Hubbard Hamiltonian impossible. Indeed, the complexity of the Hubbard problem increases exponentially with the number of sites in the chain, when it is finite, or the number of sites contained in a supercell is reproduced periodically. Different levels of approximation exist to address the problem, among which the mean-field approximation is the simplest [35]. At this level, the problem is easy to tackle, especially in the case of perfect ferromagnetic or antiferromagnetic order. More original is the study of disordered magnetic configurations of the infinite chain, which forms the main contribution of this work.

### 3.2. Energetics

The on-site elements of the TB+*U* Hamiltonian with one orbital per site are written as $\epsilon_{i,s}=\epsilon_{i}+Un_{i,-s}$, where *s* denotes spin ↑ or ↓ and −s denotes the opposite spin orientation. Let $\epsilon_{i}=\epsilon_{0}+V_{i}$, where $\epsilon_{0}$ comes from the individual atoms and $V_{i}$ is the crystal field contribution plus a possible local perturbation of the on-site elements brought about by an external perturbation. In the latter case, the one-dimensional Hubbard chain becomes spatially inhomogeneous, which introduces a further complication [36]. In this notation, the Hamiltonian is written as
(4)$$H=\sum_{i,s}\left(\epsilon_{i,s}|i,s\rangle \langle i,s|+\beta |i,s\rangle \langle i+1,s|-\frac{1}{2}Un_{i,s}n_{i,-s}\right)$$
The underlying chain model is illustrated in Figure 4.

In Equation (Equation 4), $n_{i,s}$ is the occupation of the local density of states (DOS) $\rho_{i,s}\left(E\right)$ on site *i* by an electron of spin *s*. The cohesive energy is the difference between the total energy of the condensed phase and the total energy of the same number of isolated atoms. Its expression is
(5)$$\begin{array}{cc} E_{c}= & \sum_{i}\left(\int_{-\infty }^{\epsilon_{F}}\left[\rho_{i,↓}\left(E\right)\left(E-\epsilon_{i,↓}\right)+\rho_{i,↑}\left(E\right)\left(E-\epsilon_{i,↑}\right)\right]\;dE+V_{i}\left(n_{i,↓}+n_{i,↑}\right)\right)\\ \; & +U\sum_{i}\left(n_{i,↓}n_{i,↑}-n_{0}^{2}\right)+V_{r} \end{array}$$
where $n_{0}$ is the number of electrons per spin in the isolated atoms and $V_{r}$ the total contribution of the repulsive potential between pairs of atoms in the condensed phase. The actual expression of $V_{r}$ does not matter as long as we compare the energy of the same structure with different magnetic structures. Similarly, the term involving $V_{i}$ disappears by subtraction when it involves the same set of elements $V_{i}$ in the two condensed phases that are compared.

Equation (Equation 5) is valid under the implicit assumption that each site brings $n_{0}$ electron per spin on the average. The Fermi energy $\epsilon_{F}$ follows from that condition, whisch imposes that $V_{i}$ be a constant crystal field term, the same for all the condensed phases under consideration. If a perturbation of the on-site energy is applied locally or if some local disorder is introduced, such as a Bloch wall or a small domain with reversed magnetic moment, the Fermi level $\epsilon_{F}$ remains that of the non-perturbed system. The number of electrons is not conserved and the energy modification must be computed in the grand canonical framework:(6)$$\Delta E=\Delta E_{c}+\left(\epsilon_{0}-\epsilon_{F}\right)\Delta N$$
where ΔN is the variation in the total number of electrons. The latter can be generated by the external perturbation, when there is one, or it can be accommodated by the other bands that cross the Fermi level in the real MoS${}_{2}$ nanoribbon. The term $\epsilon_{0}\Delta N$ in Equation (Equation 6) is the result of the modification in the number of electrons compared to the set of individual atoms.

### 3.3. Perfect Magnetic Structures

With the parameters of Table 2, the Stoner criterion for magnetic instability $U\rho (\epsilon_{F})>1$ is met [31]. It means that the paramagnetic chain is unstable compared to the magnetic chain. Here, $\rho \left(\epsilon_{F}\right)=2/\left[\pi W\sin\left(\pi n_{0}\right)\right]$ is the density of states per atom and per spin at the Fermi energy of the non-magnetic chain with $n_{0}$ electrons per spin and per atom, where W=4|β| is the bandwidth. Ferromagnetic and antiferromagnetic structures were considered. Self-consistent calculation of the occupation numbers of the spin-polarized densities of states were performed for the one-dimensional Hubbard model in the mean-field approximation. The results are shown in Figure 5. The energy gain per atom were evaluated for both structures with Equation (Equation 5); the results are listed in Table 3. The ferromagnetic structure is the most stable among the three structures. The calculations were performed for the band occupation $n_{0}$ = 0.65, consistent with the full electronic band structure of the MoS${}_{2}$ nanoribbon.

The widths of the spin ↑ and spin ↓ bands of the antiferromagnetic chain are small compared to the bandwidth 4β = 0.18 eV of the non-magnetic chain. The smallest value of the bandwidth is realized when $n_{0}=0.5$, where it becomes close to $4\beta^{2}/U=0.011$ eV. At half filling, the antiferromagnetic structure would be the most stable, with $\Delta E_{c}$ = −0.138 eV/atom if all parameters remain the same. As usual with the Hubbard model for half-filled band systems, the chain would then be an antiferromagnetic insulator (see also Ref. [27]).

### 3.4. Bloch Wall

The formation energy of a Bloch wall was calculated by reversing the spin orientations in half of the atomic chain. More precisely stated, the occupation numbers $n_{i,↓}$ and $n_{i,↑}$ were set to 1 and 0.3, respectively, in one half and to 0.3 and 1, respectively, in the other half. The occupation numbers of 20 sites on both sides of the Bloch wall were considered as variables. Starting from the initial values 0.3 and 1, these variables were adjusted self-consistently from the local integrated densities of states calculated using Equation ([Disp-formula FD3b-nanomaterials-13-03086]) on these sites. The Fermi energy was fixed at the value corresponding to the ferromagnetic order. After convergence of the self-consistent loop described in Section 2.2, the formation energy of the Bloch wall was calculated according to Equation (Equation 6), with the sum over *i* in Equation (Equation 5) running over 60 sites, 30 on both sides of the interface. These include the 2 × 20 sites surrounding the interface, whose occupation numbers were adjusted self-consistently, plus 10 more-distant sites with fixed bulk occupancies. The result is ΔE = 5.8 meV, in good agreement with the calculations performed for the full MoS${}_{2}$ nanoribbon [13].

Figure 6 represents the variation in the local magnetic moment calculated as the difference $n_{i,↓}-n_{i,↑}$. The bulk value is 0.7. In both domains, small damped oscillations are visible. The largest moment is realized at the interface, where it reaches 0.84. The smallest value arises right after that, 0.66. There is a small defect of electrons, ΔN = −0.21, compared to the perfect crystal.

### 3.5. Small Bloch Domains

Small magnetic domains with opposite moment were realized in a way similar to the Bloch wall. The occupation numbers $n_{i,↓}$ and $n_{i,↑}$ were set to 1 and 0.3 all along the chain except on 1, 2, 3 … 6 adjacent sites where their values were swapped. The occupation numbers on 20 sites on both sides of the center of the domain were calculated self-consistently, still keeping the Fermi energy at its bulk value. The total energy cost of the defects was calculated according to Equation (Equation 6). The results are listed in Table 4. Depending on the domain size, the formation energy varies between 7.5 and 14.9 meV. For the domains with four and more sites, ΔE is close to but not exactly equal to two times the energy of a single Bloch wall. There remains a small interaction between the walls. Figure 7 represents the local variations in the magnetic moment in the case of a single spin flip and for the domain with four reversed spins. On both sides of the domain, oscillations take place similar to Figure 6. The largest values of the moment are realized at the two walls, merely because the site occupation for spin ↑ is reduced there compared to the bulk value of 0.3, whereas the occupation of the majority spin remains close to one. The moment at the single-site domain is −0.98.

A defect in a one-dimensional chain leads to localized states above or below the electron band. This is the case with the small magnetic domains. Appendix A contains a qualitative description of the electronic structure of a single spin flip in the ferromagnetic system. The local density of states on the site where the spin is reversed is composed at 99% of two peaks, one per spin. The 1% that remains is distributed in the continuous bands. The peak at low energy, being pulled down from the upper band, is occupied by an electron with the minority spin, the other peak is not occupied (see Figure A1). When the domain is composed of two sites, there are four localized states. The lowest—actually two very closely degenerate peaks—can accommodate almost two electrons with the minority spin.

In Ref. [13], small magnetic domains on the sulfur edge of a MoS${}_{2}$ nanoribbon appeared as the consequence of having introduced local perturbations of the on-site energy. Due to its simplicity, the atomic chain model allows us to investigate this effect in a more systematic way. Similarly to in Ref. [13], we applied a Gaussian perturbation of the on-site energy; see Figure 8. The inter-atomic distance was taken as the length unit in such a way that atom *i* is located at abscissa $x_{i}=i$ along the chain. Then, the Gaussian perturbation of $\epsilon_{i}$ is
(7)$$V_{i}=V_{0}\exp\left[-{\left(i-i_{0}\right)}^{2}/2\sigma^{2}\right]$$
where σ is a dimensionless parameter. We set $i_{0}$ as either an integer number (maximum of the Gaussian on the atomic site $i_{0}$, like in Figure 8a) or as a semi-integer number (maximum of the Gaussian located halfway between two consecutive sites, like in Figure 8b). The energy of the perturbed chain was first calculated with the perfect ferromagnetic order. The energy was calculated next with the same Gaussian perturbation while reversing the spin on 1, 3, or 5 sites when $i_{0}$ was an integer and 2, 4, or 6 sites when $i_{0}$ was a semi-integer number. These initial configurations generated a magnetic domain with reversed moment on 1, 2 … 6 adjacent sites. In all cases, the initial spin configuration was symmetric with respect to the maximum of the Gaussian and kept that symmetry. The energy difference between the configurations with a reversed-moment domain and the configuration without the domain was calculated. The results are listed in Table 5. Even with a small amplitude $V_{0}$, the perturbation stabilizes small domains with reversed moments. Both parameters $V_{0}$ and σ have a strong influence on the energetics of the domains.

By increasing the amplitude $V_{0}$ at a fixed width (σ = 1.5), larger domains with reversed spins become energetically favorable compared to the ferromagnetic solution. For example, at $V_{o}$ = 0.025 eV only domains with reversed spins at up to three sites are stable (with ΔE = −0.4 meV), but for $V_{0}$ = 0.075 eV the stable domain size extends up to five sites. Similar trends can be observed by increasing the σ parameter at fixed potential value. For all the parameters explored, the most stable case is a single-site domain. However, as the last line of Table 5 indicates, domains with two or three reversed moments become competitive when the full width of the perturbation exceeds six bond lengths (σ>2.5). Consistently with the results of Ref. [13], a Gaussian perturbation centered on an atom of the chain (which represents an edge site of the MoS${}_{2}$ nanoribbon) can reverse the spin on that particular atom. In addition to the full TB+*U* calculations, our systematic study with the one-dimensional model also revealed the relation between the magnetic domains and the applied perturbation potential in a larger parameter space. If one imagines that the Gaussian perturbation mimics that produced by an STM tip, it will thereby become possible to locally change the low-temperature magnetic texture of the S edge of the nanoribbon.

## 4. Conclusions

The examination of edge magnetism within zigzag nanoribbons of monolayer MoS${}_{2}$ was undertaken with two distinct approaches: the density functional theory and a tight-binding Hamiltonian augmented with Hubbard *U* interactions. The outcomes from both methodologies consistently disclosed a notable disparity in the spin-polarization effect on an electron band crossing the Fermi level and coming from the S atoms at the S edge in comparison to other bands. This band was identified as being responsible for the biggest part of the polarization. Notably, its dispersion shape closely mirrors the energy dispersion observed in a linear chain of atoms interconnected through nearest-neighbor interactions. Exploiting this similarity, a simplified model based on an isolated atomic chain was thought up, based on the assumption that it captures the main trends of the physics of the real system. Band-structure calculations allowed us to make a one-to-one correspondence between the linear chain of the model and the two terminal sulfur atoms at the S edge of the nanoribbon. On every pair of S atoms at the edge, there is a mixing of $3p_{y}$ and $3p_{z}$ orbitals, imposed by the ribbon geometry, which are weakly bound by π interactions along the edge (the β parameter listed by Table 2).

The linear-chain model aimed at facilitating the exploration of the energy landscape for diverse spin configurations at the nanoribbon edge. With a band occupation of $2n_{0}$ = 1.3 electrons in the paramagnetic state, this model confirmed that the most stable magnetic configuration is the ferromagnetic one predicted by the full calculations. The model also accurately reproduced the energy needed to create a single Bloch wall between two domains with opposite spins, aligning well with calculations for the complete MoS${}_{2}$ nanoribbon (5.8 meV *versus* 6.5 meV). For smaller domains, we determined how the formation energy changed with domain size. Thanks to the simplicity of the atomic chain model, we could systematically investigate the effects of local changes in the on-site energy. By applying Gaussian perturbations and reversing the spins in a small portion of the chain centered at the maximum of the Gaussian profile, we calculated and compared energy differences for various configurations. The data listed in Table 5 demonstrate a lowering of the total energy of the chain perturbed by a Gaussian potential when the spin is reversed in a small region. The size of the reversed-spin domain and the energy gain depend on the Gaussian parameters. This finding agrees with a similar conclusion drawn from full TB+*U* calculations that a local Gaussian perturbation may create a small domain with reversed spins along the S edge, without destroying the ferromagnetic order in the long range [13]. We anticipate that the same one-dimensional model can be applied to study the magnetic structure of other TMDC nanoribbons, such as WS${}_{2}$, MoSe${}_{2}$, and MoTe${}_{2}$.

## Figures and Tables

**Figure 1 nanomaterials-13-03086-f001:**
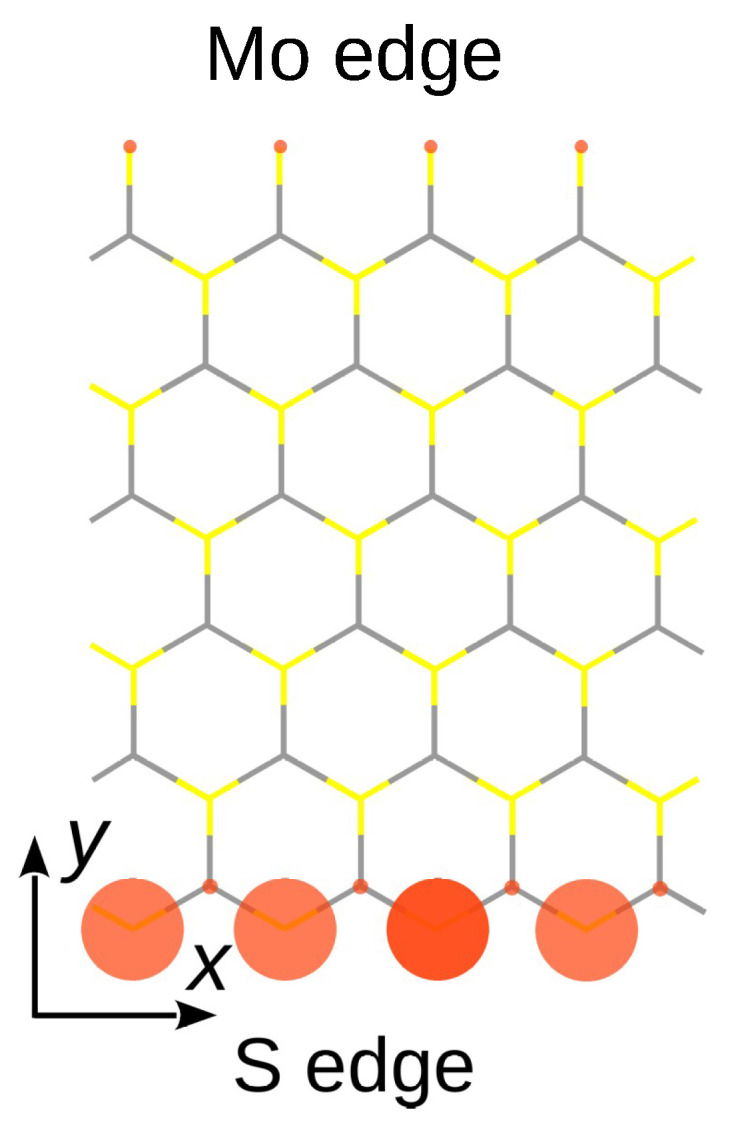
Spin polarization plots (red disks) for a zigzag MoS${}_{2}$ nanoribbon with six unit cells across thickness. The projection of the atomic structure on the Mo (x,y) mirror plane is shown by colored sticks. The Mo atoms are located at the intersection of the gray part of the sticks. The projection of the S atoms, symmetrically located above and below the Mo plane, lies at the crossing of the yellow parts. The nanoribbon is infinite in the *x* direction. The sulfur dimers are perpendicular to the plane of the drawing at the end of the topmost yellow bars.

**Figure 2 nanomaterials-13-03086-f002:**
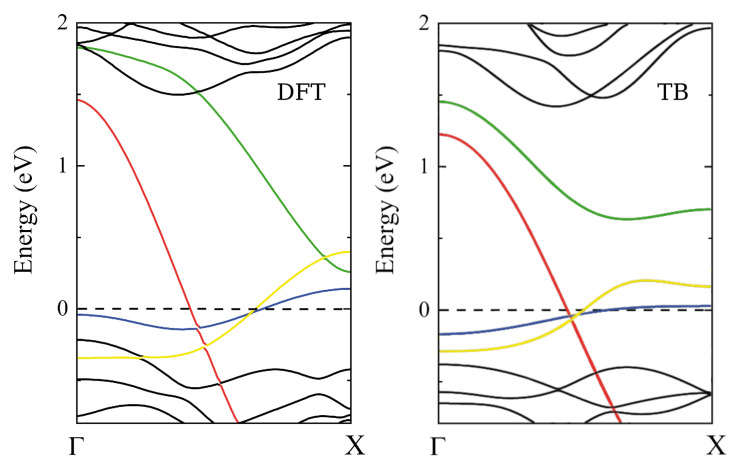
Non-spin-polarized band structure of the zigzag MoS${}_{2}$ nanoribbon with 6 unit cells across thickness. (**Left**): DFT calculations, (**Right**): TB calculations using the modified TB parameters (Table 1) for the edge atoms. The energy is measured from the Fermi level (reproduced with permission from Ref. [13]).

**Figure 3 nanomaterials-13-03086-f003:**
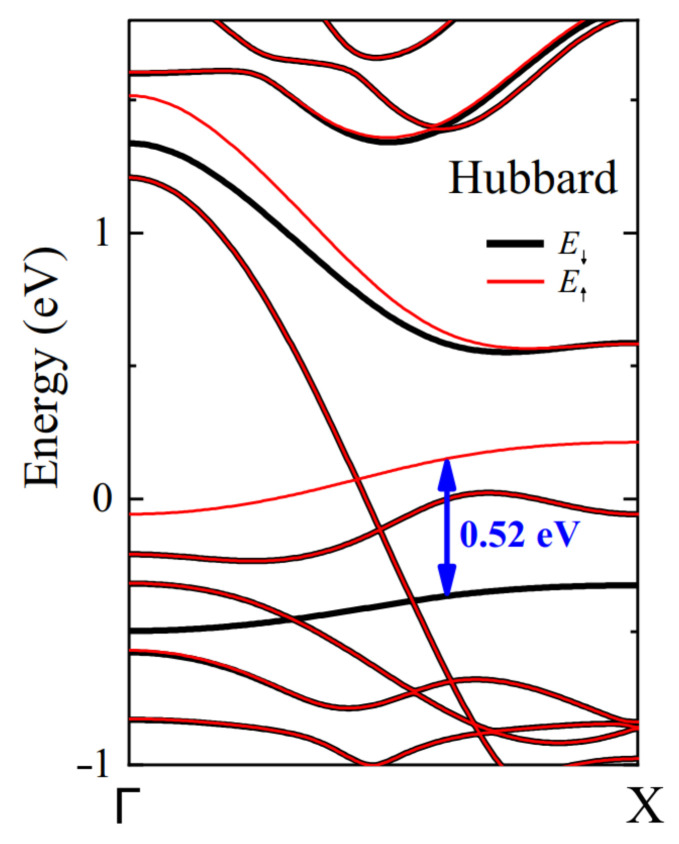
Spin-polarized tight-binding band structure of the zigzag MoS${}_{2}$ nanoribbon derived from a Hubbard model with $U_{S}$ = 1.7 eV and $U_{Mo}$ = 0.6 eV parameters. Red and black curves correspond to the up and down spins, respectively. The energy is measured from the Fermi level (reproduced with permission from Ref. [13]).

**Figure 4 nanomaterials-13-03086-f004:**
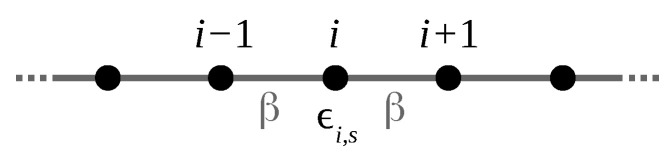
Ball-and-stick model of the linear chain described by the Hamiltonian (Equation 4).

**Figure 5 nanomaterials-13-03086-f005:**
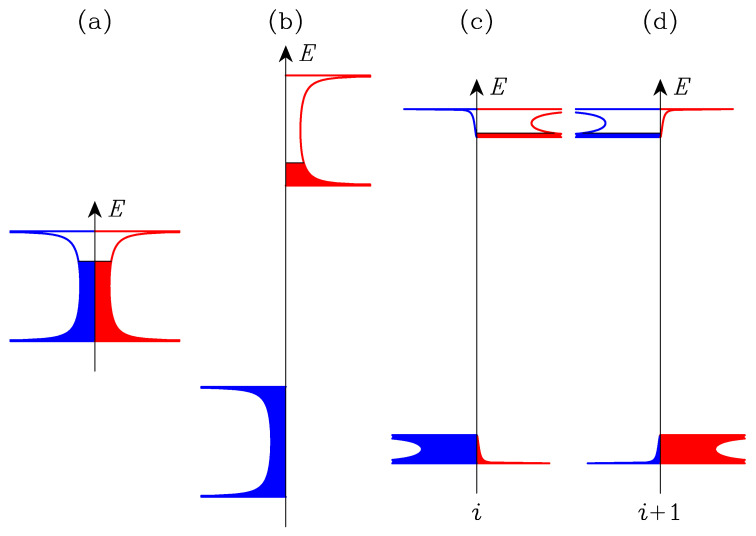
Density of states per atom and per spin of the atomic chain for three periodic structures: (**a**) non-magnetic, (**b**) ferromagnetic, and (**c**,**d**) antiferromagnetic. Spin ↓ (blue color) and spin ↑ (red color) are represented on the left and right sides of the energy axis, respectively. In the antiferromagnetic case, (**c**,**d**) display the local DOS on two successive sites *i* and i+1 along the chain. For clarity, the width of the spin ↑ and spin ↓ bands for the antiferromagnetic ordering has been enlarged by a factor of 3.

**Figure 6 nanomaterials-13-03086-f006:**
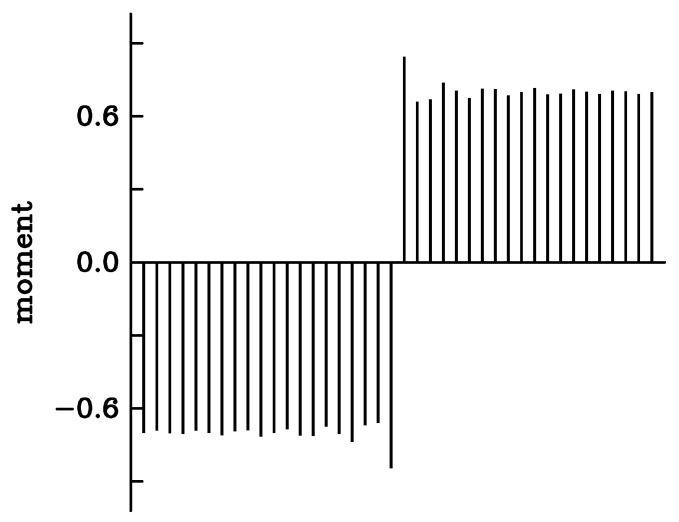
Variation in the local magnetic moment calculated along the atomic chain when reversed spins are imposed on both terminal ends of the chain.

**Figure 7 nanomaterials-13-03086-f007:**
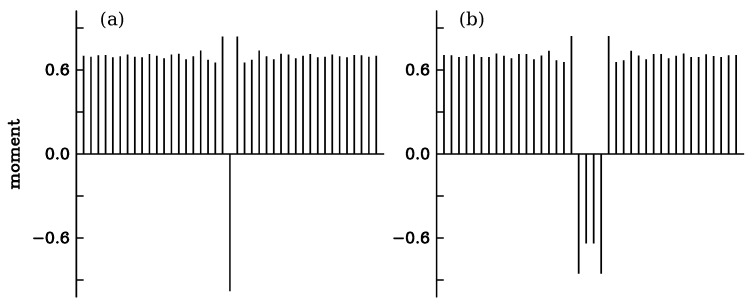
Variation in the local magnetic moment calculated along the atomic chain with (**a**) a single spin flip and (**b**) a 4-site domain of reversed spin in the ferromagnetic system.

**Figure 8 nanomaterials-13-03086-f008:**
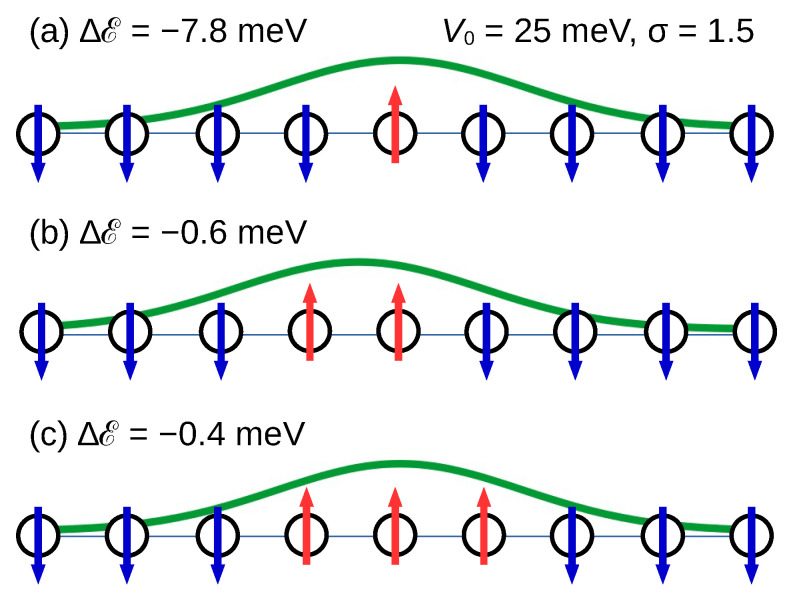
Illustration of the results listed in the first row of Table 5. The applied Gaussian perturbation (Equation (Equation 7)) is represented by a continuous green line. Its maximum is located either on top of a site (**a**,**c**) or at the center of a bond (**b**). The total energy of domains with reversed moments on (**a**) 1, (**b**) 2, and (**c**) 3 sites is indicated on the left-hand side. Negative values mean more stable configurations than the ferromagnetic chain subjected to the same Gaussian perturbation. Spins down and spins up are represented by blue and red arrows, respectively.

**Table 1 nanomaterials-13-03086-t001:** Modified tight-binding on-site energy parameters for the edge atoms in units of eV. The values correspond to the five orbitals ($4d_{xy}$, $4d_{yz}$, $4d_{xz}$, $4d_{x^{2}-y^{2}}$, $4d_{3z^{2}-r^{2}}$) for Mo atoms and three orbitals ($3p_{x}$, $3p_{y}$, $3p_{z}$) for S atoms, in the indicated order.

-	Mo Atom	S Atom	S Dimer
Mo edge	−2.03, 1.42, 1.42, −4.03, 0.51	0.28, −8.28, −12.24	−0.55, −5.28, −8.24
S edge	−2.03, 4.30, −0.80, −12.03, −2.60	−1.90, 0.18, −6.50	-

**Table 2 nanomaterials-13-03086-t002:** TB+*U* Hamiltonian parameters of the atomic chain that reproduces the dispersion of electronic states localized on the zigzag S edge of MoS${}_{2}$ nanoribbons. The energy reference is the same as for the on-site energies of Table 1.

ϵ (eV)	β (eV)	$n_{0}$	*U* (eV)	$\epsilon_{0}=\epsilon -{Un}_{0}$ (eV)
−0.8397	−0.04595	0.65	0.7428	−1.3225

**Table 3 nanomaterials-13-03086-t003:** Stabilization energy of two periodic magnetic structures of the linear chain compared to the non-magnetic case. The Hamiltonian parameters are those of Table 2.

Structure	Non-Magnetic	Ferromagnetic	Antiferromagnetic
$\Delta E_{c}$ (meV/atom)	0	−62	−47
$n_{↓}-n_{↑}$	0.00	0.70	±0.67
$\epsilon_{F}$ (eV)	−0.7980	−0.6337	−0.5874

**Table 4 nanomaterials-13-03086-t004:** Total energy cost from Equation (Equation 6) and total charge variation for small domains with reversed magnetic moment in the ferromagnetic chain. The size is the number of sites in the domain.

Size	1	2	3	4	5	6
ΔE (meV)	7.5	14.9	11.6	9.8	11.9	12.9
ΔN	−0.46	−0.75	−0.09	−0.42	−0.72	−0.01

**Table 5 nanomaterials-13-03086-t005:** Formation energy (Equation (Equation 6)) in meV for small domains with reversed magnetic moment in the ferromagnetic chain in the presence of a Gaussian perturbation with parameters $V_{0}$ and σ. The size is the number of sites in the domain and σ is given in units of the bond length.

Size	1	2	3	4	5	6
	$V_{0}$ = 0.025 eV, σ = 1.5					
ΔE (meV)	−7.8	−0.6	−0.4	+8.8	+12.4	+11.0
	$V_{0}$ = 0.05 eV, σ = 1.5					
ΔE (meV)	−11.1	−8.3	−7.4	−2.4	+1.0	+8.3
	$V_{0}$ = 0.075 eV, σ = 1.5					
ΔE (meV)	−11.3	−10.4	−9.4	−6.5	−3.0	+2.7
	$V_{0}$ = 0.05 eV, σ = 2.0					
ΔE (meV)	−11.3	−9.7	−9.3	−6.4	−4.2	+0.6
	$V_{0}$ = 0.05 eV, σ = 2.5					
ΔE (meV)	−11.4	−10.4	−10.2	−8.4	−7.1	−4.0

## Data Availability

Essential data is contained within the article in the form of tables and figures. Additional numerical data is available on request from the corresponding author.

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
