# Peer review of "Edge Magnetism in MoS2 Nanoribbons: Insights from a Simple One-Dimensional Model"

_nanomaterials, 2023, doi:10.3390/nano13243086_

Round 1

Reviewer 1 Report

Comments and Suggestions for Authors

In this manuscript, the authors studied edge magnetism in MoS2 nanoribbons using both density functional theory calculations and one-dimensional models. The edge magnetism of 2D transition-metal dichalcogenide materials sounds to be an interesting topic and can hold potential applications. A fundamental theoretical and computational study is important to understand such a phenomenon. This manuscript, however, still has significant space to be improved. Some comments are below:

i) The manuscript lacks a distinct and scientifically grounded research motivation.

ii) The manuscript is unclear regarding the documentation of experimental observations of edge magnetism in MoS2.

iii) The benefits of the tight-binding plus U approach over the current density functional theory approach are not clearly explained.

iv) It is advisable to incorporate the crystal structure model of the MoS2 nanoribbon and utilize spin density plots to clarify the origin of edge magnetism.

Author Response

See attached pdf document

Reviewer 2 Report

Comments and Suggestions for Authors

In the presented work entitled „Edge magnetism in MoS2 nanoribbons: insights from a simple one-dimensional model” by P. Castenetto et al., the Authors analyze edge states in MoS2 nanoribbons within the combination of the tight-binding, density functional theory and Hubbard model methods. The methods reveal that one band crossing the Fermi level is much strongly influenced by spin polarization than any other bands. As a result, a toy model is designed to study  the energetics of different spin configuration of the nanoribbons edge.

In general, the article is written in a well-organized manner, and I did not notice any major errors in the conducted analysis. The discussion is relatively clear and allows readers to get all the technical aspects of the presented investigations. The subject of the manuscript also appears to be timely, especially in terms of the transition metal dichalcogenide (TMDs) nanomaterials.

However, my main critique is about the employed model and, most importantly, approximations. I have a feeling that the Authors may sacrifice too much physics by not considering some important aspects of TMDs and employing simplistic toy model. In particular the Authors should provide answer to the following points in order to clarify and validated their analysis:

1.     I’m not sure if it was intended but the presented paper has a good pedagogical sound in terms of the Hubbard model. Everything is described in a clear and well-organized manner so even less experienced readers can follow-up the analysis and the theoretical model. However,I also have several points that may improve this paper even more, with the benefits for the readers. First of all, it would be instructive to describe in details the self-consistent part of calculations i.e. to explain the reads how exactly the occupations numbers of Eq. 2 are estimated. The Authors may provide step-by-step procedure for the one-dimensional model for better clarity. Secondly, I do not fully understand how the Authors determined the U value. Is it done by fitting the band structure of the Hubbard model to the DFT results? Thirdly, how the correctness of the DFT results was determined; are they compared to the previous DFT calculations?

2.     To the best of my understanding, when Authors consider non-interacting tight-binding picture (Eq. 1 and Fig.1 TB) they assume spins to be degenerated. The degeneration is lifted once the mean-field Hubbard term is introduced. However, the TMDs are well known for their strong spin-orbit coupling. If I am not missing something, the spin-orbit coupling is not considered here. But can the spin-orbit be really neglected without sacrificing important predictive features of the model. First the Authors, should present their general reasoning why the spin-orbit can be neglected and that it does not play an important role in terms of the edge states (at least in their specific analysis). Please refer to the fact that spin-orbit is not only notably influences the TMDs properties but also allows to build effects based on this interactions (refer to e.g. the spin-polarized tunneling currents in TMDs, Physical Review B 101 (11), 115423 (2020)). Second the Authors should elaborate on the possible interplay of spin-orbit and the electron-electron coupling effects. For example, as can be seen in Fig. 2 the inclusion of the Hubbard terms introduces new level around E=0 eV and changes the band gap size. The spin-splitting is also the result of the spin-orbit coupling in TMDs. Hence, what is and how we can estimate the contribution of each of the terms to be sure that we are actually talking about a given effect, not just compensating the other one.

3.     The Authors reduce the entire problem to one-dimension. However, I feel that we are losing a lot of physics here. The Authors should answer what characteristic features of the TMDs ribbons are retained in this approach and what feature are lost (and why we can neglect them). Second, once the problem is already one-dimensional what new insight, we are getting from the contacted calculations that has not be provided before during other considerations of the Hubbard chains (see e.g. Z. Physik B - Condensed Matter 49, 313-317 (1983); note that there are many other similar studies in the literature on the one-dimensional Hubbard model)? In respect to the above, I cannot see any comment on the symmetry of the employed one-dimensional model. Is it just and s-type symmetry? If yes, how does it related to the complex orbital structure of TMDs (particularly the physics of the d-orbitals; are they not important for the edge states and can be approximated with the s-symmetry? Is this not a too big approximation?)

4.     There are several minor comments. The Authors often start their sentences with word “using”. Kindly try to diversify your language for better presentation. Also, I believe that single sentence at the beginning of section 2 can be merged with the later text, so it does not feel like an unfinished paragraph. Finally, the introduction does not mention the spin-orbit coupling and its role in shaping electronic properties of the TMDs. This should be added to the introduction along with some applications of this effect for building new functionalities (e.g. like in the case of the already mentioned spin filters in point 2). This part of the text should also already include the reasoning why the spin-orbit can be neglected in the present paper.

5.     I think that conclusions also need some improvements as they are just a short, simple, and straightforward description of the obtained results. I would like to encourage the Authors to improve this section so the Readers can get some more new insight from their conclusions. Due to the character of the reported results, I believe that providing readers with the pertained perspectives would be a good idea.

Author Response

See the attached document

Round 2

Reviewer 1 Report

Comments and Suggestions for Authors

The authors have addressed my comments appropriately. I would like to recommend it for publication. 

Reviewer 2 Report

Comments and Suggestions for Authors

The Authors answered all my critique in in a satisfactory manner, therefore I recommend the present paper for publication in Nanomaterials journal.